# Glycosylated Zein Composite Nanoparticles for Efficient Delivery of Betulinic Acid: Fabrication, Characterization, and In Vitro Release Properties

**DOI:** 10.3390/foods11172589

**Published:** 2022-08-26

**Authors:** Fei Peng, Yu Jin, Kunhua Wang, Xiaojing Wang, Yaqing Xiao, Huaide Xu

**Affiliations:** 1College of Food Science and Engineering, Northwest A&F University, Yangling 712100, China; 2Food Processing Research Institute, Anhui Engineering Laboratory for Agro-Products Processing, School of Tea and Food Science & Technology, Anhui Agricultural University, Hefei 230036, China

**Keywords:** zein, betulinic acid, structural properties, nanodelivery system

## Abstract

Betulinic acid (BA) has anti-inflammatory, antioxidative stress, and antitumor activities, but BA bioavailability is low due to its poor water solubility and short half-life. This study aimed to construct a BA delivery system to improve its utilization in vitro. Glycosylated zein (G-zein) was prepared using the wet heating method, and BA-loaded zein composite nanoparticles were prepared using the antisolvent method. Compared to zein, G-zein had the advantages of higher solubility and lower surface hydrophobicity. The encapsulation efficiency of G-zein@BA reached over 80% when the BA concentration was 1 mg/mL. Compared to zein@BA nanoparticles, G-zein@BA was characterized by smaller droplets, higher encapsulation efficiency, and a more stable morphology. The sustained release and solubility of G-zein@BA nanoparticles were also superior to those of zein@BA. Compared with free BA, the dispersions of zein@BA and G-zein@BA nanoparticles in water increased 2.27- and 2.91-fold, respectively. In addition, zein@BA and G-zein@BA nanoparticles markedly inhibited the proliferation of HepG2 cells. This study provides new insights into the structural properties and antitumor activity of BA composite nanoparticles to aid in the development of zein particles as functional materials to deliver bioactive compounds.

## 1. Introduction

Betulinic acid (BA) is a lupane-type pentaloidal triterpenoid that is widely distributed in the bark, leaves, flowers, and fruits of plants, such as birch, rosemary, perilla, loquat, clove, coffee, jujube, and apple [1]. Previous studies have shown that BA exerts various biological activities, such as antitumor, liver protection, anti-inflammatory, antihuman immunodeficiency virus, antioxidative stress, antiatherogenic, and antiobesity [2]. Among them, the antitumor activity of BA has been extensively studied, for example, in liver cancer [3]. However, the poor water solubility (21 μg/mL) and low bioavailability of BA due to its short half-life greatly limit its clinical application [4]. To simultaneously address the problems of solubility and poor pharmacokinetics, nanotechnology has become an important means of improving the biological activity of BA [5].

As a prolamin protein extracted from corn, zein is a raw food material generally recognized as safe (GRAS) by the Food and Drug Administration of the United States [6]. Zein produces microspheres via the antisolvent precipitation technique, and structurally has excellent embedding morphology, making zein a good carrier material [7]. Due to its excellent hydrophobicity, biodegradability, biological adhesion, and economic characteristics, zein has been used as a functional component in delivery systems [8]. Importantly, glycosylated proteins exhibit improved dispersion, solubility, and bioactivity, which could improve the particle distribution, stability, and encapsulation efficiency of active components [9].

Traditional BA usually carries nanotransport to achieve solubilization by reducing the carrier particle size, changing the pH, and using cosolvents and surfactants, which might be detrimental to health due to the nonspecific tissue distribution of drugs [10]. Zein and bioactive compounds can generate complexes through noncovalent interactions [11]. Zein easily forms spherical colloidal nanoparticles through self-assembly. Zein and hydrophobic functional components have been dissolved in ethanol, and the functional components could be wrapped in zein nanoparticles using the antisolvent coprecipitation method [12]. Zein is suitable as a delivery carrier to improve the bioactivity of BA; however, there is little research on constructing glycosylated zein as a nanocarrier for BA encapsulation and delivery.

In this study, glycosylated zein (G-zein) and BA-loaded nanoparticles (zein@BA and G-zein@BA) were prepared. The interface, spectral properties, amino acid composition, and molecular weight of G-zein were then analyzed. Structural properties, including the action force, crystal type, apparent morphology, stability, slow-release, and water dispersion of BA-loaded nanoparticles, were analyzed. Furthermore, we investigated the antiproliferative activity of BA-loaded nanoparticles in HepG2 tumor cells. Most importantly, compared to zein@BA nanoparticles, G-zein@BA had the advantages of smaller droplets, higher encapsulation efficiency, a more stable morphology, and better-sustained release and solubility. Our study might provide a better scientific understanding of the BA delivery system wrapped with plant protein.

## 2. Materials and Methods

### 2.1. Materials

BA (purity ≥ 98%) was purchased from Nanjing Chunqiu Biological Engineering Co., Ltd. (Nanjing, China). Zein (protein content 92%) was provided by Sigma–Aldrich, Inc. (St. Louis, MO, USA). The human hepatocellular carcinoma HepG2 cell line was obtained from Prof. Wang [13,14]. Phthalaldehyde, coomassie brilliant blue G-250, and potassium bromide (spectral grade) were purchased from Sinopharm Chemical Reagent Co., Ltd. (Shanghai, China). Standard protein (14.4–97.4 kDa), horseradish peroxidase, and β- mercaptoethanol were purchased from Thermor Science & Technologies Co., Ltd. (Xi’an, China). Additionally, 8-Amino-1-naphthalene sulfonic acid (ANS) was purchased from J&K Scientific Co., Ltd. (Beijing, China). Folin-phenol reagent, dialysis bag (molecular interception capacity of 8000 kDa), and cell counting kit-8 (CCK-8 kit) were purchased from Solarbio Biotechnology Co., Ltd. (Beijing, China). All chemicals, reagents, and solvents were analytically pure or of a higher grade.

### 2.2. Preparation of Zein Glycosylated Products

Zein-glycosylated products were prepared using a previously described method [15]. Briefly, zein powder was dissolved in 0.2 mol/L KCl-NaOH buffer (pH = 13.0) under magnetic stirring until thoroughly dissolved. Under constant magnetic stirring and heating, sugars were added to the solution to mix zein and sugar, undergoing the Maillard reaction. Then, the mixture was placed on ice to stop the reaction, the supernatant was separated by centrifugation, and the pH value was adjusted to 7.0 using 1 mol/L HCl. Zein glycosylated product was obtained by freeze-drying powder (FD5-series, Gold-SIM International, Beijing, China) after dialysis with a molecular interception capacity of 8000 kDa for 48 h. To improve the browning degree of zein-glycosylated product, four primary factors were assessed based on the index of sugar grafting degree. A single-factor experiment was performed on the sugar types (glucose, fructose, xylose, galactose, lactose, maltose, dextran, and maltodextrin), zein concentration (1%, 2%, and 3%), reaction time (1, 2, 3, and 4 h), and the mass ratio of zein to sugar (2:1, 1:1, and 1:2). The zein glycosylation product with the highest degree of grafting was named G-zein.

### 2.3. Characterization of Zein and G-Zein

The grafting degree of zein-glycosylation products was determined using the ortho-phthalaldehyde (OPA) method [16]. In the dark, OPA powder (80 mg) was dissolved in 95% ethanol (2 mL, *v/v*) by ultrasound. The OPA reagent (100 mL) was obtained by mixing 2 mL of OPA solution, 50 mL of sodium tetraborate buffer (10 mmol/L, pH 9.7), 5 mL of SDS solution (20%, *w/v*), 200 μL of β-mercaptoethanol, and distilled water. Zein and G-zein solutions (2 mg/mL, 200 μL) were prepared in 75% ethanol (*v/v*) and added to 4 mL of OPA reagent. After 10 min of reaction, the absorbance was examined at 340 nm. Zein was the control group, and G-zein was the experimental group. The formula for the grafting degree is as follows. In this formula, *A_C_* represents the absorbance of the control group, and *A_S_* represents the absorbance of the experimental group.
(1)$$Grafting\;degree\;\%=\frac{A_{C}-A_{S}}{A_{C}}\times 100\%$$

Zein and G-zein solutions (1% protein concentration, *w/v*) were prepared in 75% ethanol (*v/v*) under magnetic stirring for 1 h. The pH values were adjusted to 2, 4, 6, 8, 10, and 12 using 0.1 mol/L HCl and NaOH. The supernatant was centrifuged at 12,000× *g* for 30 min. The protein content of the supernatant was determined using a Folin-phenol reagent using a microplate fluorometer (Fluoroskan, Thermo, Waltham, MA, USA). The solubility of zein and G-zein was the mass ratio of protein content in the supernatant to the total protein in the sample.

Zein and G-zein powders were dissolved in 0.02%, 0.04%, 0.06%, 0.08%, and 0.1% (*w/v*) of 0.01 mol/L phosphate buffer (PBS) at pH 7.0. Next, 8-Amino-1-naphthalene sulfonic acid (ANS) solution (8 mmol/L, 20 μL) was added to the zein solution (4 mL) at different concentrations, and the fluorescence intensity was measured using a fluorescence spectrophotometer (LS55, PerkinElmer, Waltham, MA, USA) immediately after shaking. The excitation wavelength was 390 nm, the absorption wavelength was 470 nm, and the concentration of the zein solution without ANS was used as a blank control. The surface hydrophobicity (*H*_0_) of protein molecules is the slope of the protein concentration-fluorescence intensity curve fitted using the least squares method [17].

The amino acid contents of zein and G-zein were examined using the phenyl isothiocyanate precolumn derivatization method [18]. Zein and G-zein (0.10 g) were hydrolyzed using HCl (5 mL, 6 mol/L) in an ampoule at 110 °C for 24 h. Then, the amino acid composition and content were assessed using high-performance liquid chromatography (HPLC, HP1100, Agilent, Santa Clara, CA, USA).

Zein and G-zein powder (1 mg) and potassium bromide powder (100 mg) were thoroughly ground and mixed and then pressed into a thin film using a tablet press. The potassium bromide tablet was a blank control, and the scanning spectrum range was 400–4000 cm^−1^. The infrared spectra were examined using Fourier transform infrared spectroscopy (FTIR, Spectrometer 100, Bruker, Germany) [19]. OMNIC 8.0 software was used for data analysis.

Zein and G-zein were dissolved in 70% ethanol (*v/v*) with a protein concentration of 0.1 mg/mL. The scanning wavelength of circular dichroism was 190–250 nm, the scanning rate was 100 nm/min, the optical path of the colorimeter was 1 mm, and the scanning step was 1 nm [20]. Protein secondary structures, including α-helices, β-sheets, β-turns, and unordered coils, were predicted. The molar ellipticity of zein and G-zein was calculated using the following formula. In this formula, [*θ*] represents molar ellipticity (deg·cm^2^·dmol^−1^), *X* represents the millimeter signal measured by a circular dichrometer, *M* represents the average molecular weight of amino acid residues (zein was 110), *L* represents the colorimetric optical path, and *C* represents the protein concentration (mg/mL).
(2)$$\left[\theta \right]=\frac{100\times X\times M}{L\times C}$$

### 2.4. Electrophoresis of Zein and G-Zein

The molecular weight and protein subunits of zein and G-zein were analyzed using dodecyl sulfate and sodium salt-polyacrylamide gel electrophoresis (SDS–PAGE) [21]. Zein and G-zein were dissolved in a protein-loading buffer containing β-mercaptoethanol and then boiled. The concentrations of the separated gel and concentrated gel were 12% and 5%, respectively. The standard protein marker was 14.4–97.4 kDa. Protein samples (10 μg) were separated by SDS–PAGE. The electrophoretic gel was then soaked in 0.25% Coomassie blue G-250 for 2 h, followed by decolorization with 10% methanol and 5% glacial acetic acid solution. The Schiff reagent method was used for sugar staining with horseradish peroxidase as the glycoprotein marker.

### 2.5. Preparation of BA-Loaded Zein/G-Zein Nanoparticles

BA-loaded zein and G-zein nanoparticles were prepared using the antisolvent method [22]. Zein and G-zein (2 g) were dissolved in 70% ethanol (*v/v*, 100 mL) and stirred magnetically at 600 rpm for 2 h at 25 °C. Then, BA was added to the protein ethanol aqueous solution to create BA concentrations of 1, 1.5, and 2 mg/mL and stirred magnetically at 600 rpm for 3 h at 25 °C. The mixtures (20 mL) were placed for 1 h, then added to the 60 mL aqueous solution (pH = 4) and stirred magnetically at 600 rpm for 30 min at 25 °C. Vacuum rotary evaporation was performed (45 °C; −0.1 MPa) to obtain the BA-loaded nanoparticles, zein@BA and G-zein@BA.

### 2.6. Characterization of BA-Loaded Zein/G-Zein Nanoparticles

#### 2.6.1. Particle Size, Polydispersity Index (PDI), and Zeta-Potential

The particle size, polydispersity index (PDI), and zeta-potential of BA-loaded nanoparticles were measured using a Zetasizer analyzer (ZEN3600, Malvern, UK). Samples were diluted and balanced for 1 min to avoid the multilight scattering effect of nanoparticles. For particle sizing, the medium viscosity is 0.8872, the refractive index is 1.33, and the dilution factor is 100-fold. A low PDI indicates that the particle distribution is uniform, and the sample is stable. The cumulative mean diameter was measured with the Stokes–Einstein equation, and the zeta-potential was measured in Smoluchowski mode at 25 °C [23].

#### 2.6.2. Encapsulation Efficiency

The encapsulation efficiency of BA was measured using the ultrafiltration centrifugation method. Zein@BA and G-zein@BA nanoparticles (1 mL) were added into an ultrafiltration tube with a molecular weight of 10 kDa and centrifuged at 4000 rpm for 30 min. The filtrate containing the membrane was removed, and then methanol (1 mL) was added to the ultrafiltration tube for further centrifugal extraction. The combined filtrate was filtered through a 0.22 μM organic filter membrane and assessed by HPLC. This assay was performed on a Waters Symmetry C18 column (5 µm, 250 mm × 4.6 mm) with a mobile phase consisting of acetonitrile:water (75:25, *v/v*) at a flow rate of 1.0 mL/min. The wavelength was 210 nm, and the column temperature was maintained at 30 °C. The encapsulation efficiency was the mass ratio of BA-loaded nanoparticles to total BA.
(3)$$Encapsulation\;efficiency\;\%=\frac{mass\;of\;BA\;in\;nanoparticles}{the\;total\;mass\;of\;BA\;used}\times 100\%$$

#### 2.6.3. FTIR

Zein, G-zein, BA, zein@BA, and G-zein@BA powder (1 mg) and potassium bromide powder (50 mg) were thoroughly ground and mixed and then pressed into a thin film using a tablet press. The potassium bromide tablet served as a blank control, and the scanning spectrum range was 400–4000 cm^−1^. FTIR spectral curves were analyzed using OMNIC 8.0 software.

#### 2.6.4. Transmission Electron Microscopy (TEM)

Nanoparticle suspensions were added to a copper net containing a carbon support film. After natural drying overnight, microscopic images of zein@BA and G-zein@BA nanoparticles were obtained with transmission electron microscopy (TEM, JEM-1230, JEOL, Tokyo, Japan) at 100 kV.

#### 2.6.5. X-ray Diffractometer

Nanoparticles were freeze-dried into powder and placed into an X-ray diffractometer (Ultimate IV, Rigaku, Japan) to analyze crystal morphology. The diffraction angle was 2θ, the scanning range was 10–60°, the velocity was 5°/min, and the current was 40 mA.

### 2.7. Stability of BA-Loaded Zein/G-Zein Nanoparticles

The pH of the zein@BA and G-zein@BA nanoparticles was adjusted to 2, 4, 6, 8, 10, and 12 using HCl or NaOH (1 mol/L or 0.1 mol/L, respectively), and morphological changes were observed. In addition, NaCl was added to the nanoparticles to final concentrations of 0, 10, 20, 30, 40, and 50 mM, and morphological changes were observed. The particle size, PDI, and zeta-potential were measured.

### 2.8. In Vitro Sustained Release and Water Dispersibility of BA-Loaded Zein/G-Zein Nanoparticles

Zein@BA, G-zein@BA, and free BA ethanol solution (5 mL) were added to a dialysis bag with a retained molecular weight of 14 kDa, soaked in 25 mL PBS (pH = 7.4) containing 0.2% Tween-80, and shaken at 100 rpm at 37 °C. The solution outside the dialysis bag was removed at 0, 1, 4, 6, 12, 24, 30, 36, 48, 54, 60, and 72 h and supplemented with the blank buffer at the same volume. The cumulative release levels of BA were measured by HPLC.

To examine the embedding effect of nanoparticles on the water dispersibility of BA, the above suspensions of zein@BA, G-zein@BA, and free BA were centrifuged at 5000 rpm for 10 min, and the supernatant was filtered through a 0.22 μm filter membrane to remove the insoluble BA. The filtrate (1 mL) was vortexed in 4 mL of ethanol, and 15 mL of methanol was added to the mixture to extract BA. The level of BA in the extract solution was determined by HPLC.

### 2.9. Cytotoxicity of BA-Loaded Zein/G-Zein Nanoparticles to HepG2 Cells

HepG2 cells were cultured at 37 °C in a humidified atmosphere containing 5% CO_2_. The culture medium contained DMEM (high D-glucose) (HyClone, Logan, UT, USA), 100 U/mL penicillin/streptomycin (Harbin Pharmaceutical Group, Harbin, China), and 10% fetal bovine serum (Gibco, Gaithersburg, MD, USA). HepG2 cells were seeded into 96-well plates at a density of 1 × 10^4^ cells per well and then washed with PBS. BA-loaded nanoparticles were dissolved in DMSO to a certain concentration according to the BA level. Then, the cells were treated with blank nanoparticles, zein@BA, G-zein@BA, and free BA ethanol solution (100 μL) at different concentrations. Cells were cultured for 24 h or 48 h, and at least 6 parallel experiments were performed in each group. The effect of BA-loaded nanoparticles on HepG2 proliferation was examined using a CCK-8 kit (Solarbio, Beijing, China). Cells were incubated for 2 h with a CCK-8 solution (10 μL), and the absorbance was measured at 450 nm using a microplate reader. The cell viability of HepG2 was measured using the following formula. In this formula, *A_S_* represents absorbance of wells with cells, CCK-8 solution, and BA nanoparticles, *A*_0_ represents absorbance of wells with medium and CCK-8 solution but without cells, *A_C_* represents absorbance of wells with cells, CCK-8 solution but without BA nanoparticles.
(4)$$Cell\;Viability\;\%=\frac{A_{S}-A_{0}}{A_{C}-A_{0}}\times 100\%$$

### 2.10. Statistical Analysis

All measurements were performed in triplicate, and the results are shown as the mean ± standard deviation (SD). Data were processed using SPSS 19.0 (IBM, Chicago, IL, USA). Statistical comparisons were performed using one-way ANOVA with Tukey’s test. Significant differences were declared at * *p* < 0.05.

## 3. Results and Discussion

### 3.1. The Effect of Sugar Type, Zein Concentration, the Ratio of Sugar to Zein, and Reaction Time on the Grafting Degree of G-Zein

To obtain the zein-glycation product with the highest grafting degree, a single-factor experiment was performed by adjusting sugar type, zein concentration, the ratio of sugar to zein, and reaction time. As shown in Figure 1A, compared to disaccharides and polysaccharides, such as lactose, maltose, dextran, and maltodextrin, monosaccharides exhibited a stronger browning reaction with zein, especially xylose and glucose, whose grafting degree reached approximately 20%. The solubility of monosaccharides was better in strong alkaline solutions, and the contact between monosaccharides and zein was more sufficient and intense, so the degree of graft reaction was deeper. Compared to 1% zein, 2% or 3% zein decreased the sugar grafting degree (Figure 1B), indicating that the Maillard reaction branching chain on zein was limited. There was no significant difference in the degree of sugar grafting when the ratio of zein to glucose was changed (Figure 1C). In addition, the grafting degree significantly increased after the 2 h reaction, but there was no significant increase after 3 h or 4 h (Figure 1D), indicating that the browning reaction between zein and glucose was completed within 2 h. Therefore, glucose was selected, and a Maillard reaction was conducted to prepare G-zein under conditions of 1% zein. The ratio of zein to glucose was 2:1, with a 2 h reaction time. The grafting degree of G-zein reached approximately 22.8%.

### 3.2. Interface Characteristics of Zein and G-Zein

Zein, a proline-rich amphiphilic protein, can change its interfacial properties in response to glycosylation [24]. Figure 2A shows that the minimum solubility of zein was only 1.87% at pH = 6, while the maximum solubility was 29.47% at pH = 12, indicating that zein was more soluble under strongly alkaline conditions. The isoelectric point of zein was at approximately pH = 6, which is consistent with the report that the isoelectric point was pH = 6.2 [25]. Compared to zein, the solubility of G-zein increased significantly at different pH values (*p* < 0.05). The minimum solubility of G-zein was 11.46% at pH = 4, and the maximum solubility was 52.95% at pH = 12. Glycosylation caused the isoelectric point of zein to move toward an acidic pH = 4. The combination of glucose and amino acid residues on the zein branch chain might make G-zein have more hydrophilic groups, for example, hydroxyl groups. In addition, the steric hindrance introduced by glucose may prevent protein aggregation in G-zein, improving the solubility of G-zein [26].

The surface hydrophobicity of a protein has an important effect on its structural stability and interfacial properties. Compared to zein, the surface hydrophobicity of G-zein was significantly reduced (*p* < 0.05; Figure 2B), which may be due to the increase in hydrophilic groups on the zein side chain by the graft of glucose [27]. In addition, expansion of the zein structure leads to the exposure of hydrophobic amino acid residues originally hidden in the tertiary structure of zein and decreases the surface hydrophobicity. Similarly, maltodextrin grafted onto soybean protein significantly reduced its surface hydrophobicity [28]. The surface hydrophobicity of the SPI-dextran conjugate decreased with an increasing grafting degree. The glycosylation reaction between soybean and gum acacia decreased the hydrophobicity of soybean protein [29], which indicated that sugar reduced the hydrophobicity of the protein.

As shown in Figure 2C, the amino acid compositions of zein and G-zein were similar, indicating that glycosylation did not change the amino acid composition. The results showed that levels of glutamate (Glu) were the highest, followed by isoleucine (Leu) and proline (Pro). These three amino acids accounted for more than 70% of total amino acids, which is consistent with other research [7]. Compared to zein, the hydrophobic amino acids (Leu and Pro) decreased while hydrophilic amino acids (Glu) increased to some extent in G-zein, which may lead to enhanced hydrophilicity.

### 3.3. Spectroscopic Analysis of Zein and G-Zein

FTIR is based on the different radiation absorption spectra caused by the vibration of different atoms and reflects changes in protein intermolecular force and structure. Figure 2D shows the FTIR spectra of zein and G-zein. Zein displayed three characteristic peaks of amide I (1600–1700 cm^−1^), amide II (1530–1550 cm^−1^), and amide III (1240–1450 cm^−1^), which were also observed in the FTIR spectrum of corn cob [30]. Compared to zein, the absorption intensity of G-zein at 1409 cm^−1^ and 1694 cm^−1^ was significantly increased, which was caused by the C=O stretching vibration in the amide I band, C-N stretching vibration, and N-H deformation vibration in the amide III bands. After glycation of soybean protein with carboxymethyl cellulose, the absorption intensity of the amide I band and amide III bands changed, which may be caused by intermediate products, such as the Schiff base generated in the glycosylation process, which enhanced the absorption peak [31]. G-zein had a significant absorption peak in the wavelength range of 1000–1260 cm^−1^, which was caused by a stretching vibration of the C-O-C glycosidic bonds in sugar molecules. The stretching vibration of free -OH in the sugar molecule increased the absorption strength at 3396 cm^−1^, indicating that zein forms a complex with glucose through a covalent bond. In addition, the absorption peaks of G-zein at 545 cm^−1^, 860 cm^−1^, and 2928 cm^−1^ all become stronger, which may be caused by the antisymmetric stretching vibration of C-H (SP^3^) in the CH_3_ group [32], further confirming that zein forms a covalent complex with glucose.

The circular dichroism spectra of zein and G-zein exhibited positive peaks at 195 nm and negative peaks at 205–230 nm (Figure 2E). Zein and G-zein were consistent with the spectral characteristics of general proteins, with positive peaks at 195 nm and negative grooves at 205–230 nm. As shown in Figure 2F, the secondary protein structure was analyzed using Dichrowb online software. After glycation, the α-helix of zein decreased, the β-sheet significantly increased (*p* < 0.05), and the content of β-turns and unordered coils showed no significant difference. Due to the grafting of sugar molecules on zein, the secondary structure of the protein changes, and the structure expands [20]. The α-helix is tight, stable, and has no cavities, while the β-sheet and β-turn are relaxed and flexible. The ductility and flexibility of G-zein are significantly enhanced, indicating new functional properties.

### 3.4. SDS–PAGE of Zein and G-Zein

Coomassie blue staining results revealed that glycosylation did not change the intermolecular structure of zein (Figure 3A). Zein containing α-zein, β-zein, and γ-zein subunits covalently reacted with only a small amount of glucose [33]. In addition, the molecular weight of glucose is only approximately 180, so the reaction site and quantitative difference between zein and G-zein could not be analyzed. Schiff reacted with sugar molecules in G-zein to stain glycoproteins, and the positive control was horseradish peroxidase (Figure 3B). The color reaction occurred on α-zein and γ-zein bands, indicating that new glycoproteins were generated, further confirming that zein combines with glucose to produce trace-glycation products [34]. These results demonstrate the successful preparation of G-zein, which has the advantage of higher solubility and lower surface hydrophobicity than zein and has the potential to construct a BA delivery system.

### 3.5. Particle Size, Zeta-Potential, and Encapsulation Efficiency of BA-Loaded Nanoparticles

Zein ethanol solution containing BA was directly injected into distilled water at a specific pH using the reverse solvent precipitation method to prepare zein@BA and G-zein@BA nanoparticles. Different from the traditional reverse solvent method, the protein and BA were dissolved in 70% ethanol (*v/v*) in our study. As shown in Table 1, with the increase in BA concentration, the particle size of zein@BA and G-zein@BA increased from 139 nm and 168 nm to 326 nm and 484 nm, respectively, while PDI was at a high level, being greater than or equal to 0.37. In addition, the particle size of G-zein@BA was always lower than that of zein@BA at the same concentration of BA. Particle size is a very critical attribute of nanocarriers, which affects stability, encapsulation efficiency, drug release profile, biodistribution, mucoadhesion, and cellular uptake [35]. Due to the enhancement of the interaction between zein and BA with increasing BA concentration, particle aggregation, instability, and poor dispersion of the system led to high PDI. To our knowledge, PDI is a number calculated from a two-parameter fit to the cumulant analysis. PDI values smaller than 0.7 indicate that the sample is suitable to be analyzed with the dynamic light scattering technique [36]. While the introduction of the sugar group in G-zein changes the interaction between protein and BA, the reverse micellar or vesicle structure with G-zein as the core may be formed, resulting in a reduction in particle size.

G-zein has a significant effect on the surface electrical properties of BA-loaded nanoparticles. Due to the ionization of amino acids, the zeta-potential of the zein@BA nanoparticles at pH = 4 was approximately +32 mV. After glycosylation of zein, the zeta-potential of G-zein@BA rapidly changed to less than −20 mV. The isoelectric points of G-zein and zein were different, and G-zein itself had a large negative charge, which could adsorb onto the surface of BA-loaded nanoparticles, making them negatively charged. After glycosylation, the zein protein structure was loose and disordered, increasing the shear plane on the surface of the nanoparticles and resulting in potential reduction. The shear plane is the distance between the particles, and these surfaces remain strongly attached to any counterions when the particles are moving in the electric field. The zeta-potential of nanoparticles is the total charge that the nanovesicle obtains in a specific environment or suspension medium. The net charge on the surface of the nanoparticles affects the ion distribution in the surrounding area, enhancing the counter-ion concentration near the surface [37]. The zeta-potential of zein nanoparticles decreased after the addition of Tween 80, which was primarily attributed to the fact that Tween 80 with a small amount of negative charge could adsorb onto zein nanoparticles with a positive charge at pH = 4.0, neutralizing part of the charge [38].

With increasing BA concentration, the encapsulation efficiency of BA in zein@BA and G-zein@BA decreased from 83% and 89% to 40% and 49%, respectively. At the same concentration as BA, the encapsulation efficiency of G-zein@BA was always higher than that of zein@BA. BA bound to the hydrophobic microregion of the protein through electrostatic and hydrophobic interactions, so that it could be effectively encapsulated in nanoparticles. With the increase of BA concentration, the hydrophobic binding site of zein/G-zein could not accommodate more BA binding, so the encapsulation efficiency decreased. The interaction force between BA and protein was limited, and as the concentration of BA increases, the protein cannot produce enough electrostatic or hydrophobic interactions with BA, resulting in particle aggregation and reduced encapsulation efficiency [39]. However, the interaction between G-zein and BA is stronger than that of zein, so the encapsulation efficiency of G-zein@BA was always higher than that of zein@BA at the same BA concentration.

### 3.6. Morphology and Structural Characterization of BA-Loaded Nanoparticles

The chemical bonds and interactions of zein@BA and G-zein@BA nanoparticles were characterized using FTIR (Figure 4A). Compared to the infrared spectra of zein, the amide band of G-zein changed, and the absorption intensity of G-zein was significantly increased at 1409 cm^−1^ and 1694 cm^−1^. Zein and glucose formed a complex through a covalent bond, which enhanced the electrostatic adsorption and the binding rate of G-zein to BA [40]. The characteristic peaks of BA are located at 541.99 cm^−1^ (O=C-O stretching vibration), 792.74 cm^−1^ (C-C stretching vibration), and 885.33 cm^−1^ (C-C stretching vibration) [41]. The absorption peaks of the amide I and amide II bands of zein nanoparticles shifted from 1661 cm^−1^ and 1517 cm^−1^ to 1659 cm^−1^ and 1535 cm^−1^, respectively, after glycosylation of BA with zein. Therefore, there is an electrostatic interaction between zein and BA. Compared to zein nanoparticles, the absorption peaks of the amide I band and amide II bands of zein and α-tocopherol composite nanoparticles shifted from 1664 and 1550 cm^−1^ to 1660 and 1547 cm^−1^, respectively, indicating electrostatic interactions between zein and α-tocopherol. In addition, both zein and BA are hydrophobic compounds, and hydrophobic interactions are another force promoting the formation of proteins and BA-loaded nanoparticles.

The crystalline state of a substance is related to its solubility. In general, crystalline solids are more soluble than amorphous solids. The spatial distribution, orientation, and intensity of diffraction lines are closely related to the crystal structure, so X-ray diffractometry is used to examine the crystal structure. Figure 4B shows the XRD patterns of zein@BA and G-zein@BA. BA had a specific crystal morphology, and a strong characteristic diffraction peak was observed between 10° and 20°. The diffraction intensity was the highest at 13.611°, 14.288°, and 19.113°. Distinct from BA, zein, and G-zein displayed two wide peaks between 10–15° and 25–35°, both of which existed in an amorphous state, and glycosylation did not change the crystal state. After embedding BA, characteristic peaks of zein@BA and G-zein@BA were not observed at diffraction angles of 2θ or 10–20°, indicating that BA was encapsulated in the nanoparticles [42]. Figure 4C shows the morphology under a transmission electron microscope. Zein and G-zein particles were spherical structures with smooth surfaces and uniform particle sizes. After embedding BA, the nanoparticles were still spherical. Compared to zein@BA, the particle size of G-zein@BA was smaller, and the binding of G-zein to BA was closer, consistent with the particle size of nanoparticles.

### 3.7. Stability of BA-Loaded Nanoparticles

The colloid delivery system will experience a variety of pH and electrolyte conditions in food industry products or in the human gastrointestinal tract [39]. It is of great significance to determine the influence of pH and ionic strength on the physical properties and stability of the colloid delivery system. As shown in Figure 5A, the particle size of zein@BA was small at pH = 2, 3, and 4, but increased at pH = 5. The dispersion formed a layer of sediment at the bottom of the glass bottle, and the supernatant was cloudy at pH = 6, which was attributed to the isoelectric point of zein being pH = 6.2. Due to the fact that the isoelectric point of zein was at approximately pH = 6, protein aggregation and precipitation occurred under this pH value condition, and the particle size of zein@BA nanoparticles increased suddenly. Compared to zein@BA, the G-zein@BA dispersion was stable with no significant change in particle size, primarily because the isoelectric point of G-zein moved in the acidic direction [43]. With increasing pH, the G-zein dispersion improved, and the combination of G-zein and BA was more solid and stable. G-zein@BA had a negative charge under different pH conditions, while zein@BA had a positive charge to the left and a negative charge to the right of the isoelectric point (Figure 5B). Zeta-potential varies with pH and becomes more positive and negative in magnitude with acidic and basic pH values, respectively [44]. Therefore, titration curves of the zeta-potential against different pH values are often produced, which helps to determine the isoelectric point (pH value) where the zeta-potential becomes zero [45].

As shown in Figure 5C, zein@BA was sensitive to salt concentration. At a low salt concentration of 20 mM, particles gathered rapidly and formed a layer of white precipitate at the bottom of the tube. As the salt concentration increased, the sediment accumulation layer thickened, and the supernatant became clear. This result is consistent with previous research showing that zein nanoparticles are unstable in NaCl solutions [46]. NaCl enhanced the electrostatic shielding effect and reduced the electrostatic repulsion between nanoparticles, so gravitational effects, such as van der Waals forces and hydrophobic effects, overcame the electrostatic repulsion and led to the aggregation of nanoparticles. Compared to that of zein@BA, the particle size of the G-zein@BA dispersion increased with increasing salt concentration until a precipitate was produced. However, the particle size of the G-zein@BA dispersion was always smaller than that of zein@BA at multiple salt concentrations (Figure 5D), indicating that glycosylation only alleviates the electrostatic shielding effect of ions on BA-loaded nanoparticles. Zein@BA and G-zein@BA have positive and negative charges at different ionic concentrations, respectively, and the absolute value of charge tended toward zero with increasing ionic concentration, indicating that the interaction between BA and protein was completely dissolved by ions. The zeta-potential decreases with increasing ionic strength, and the ion valency is also important in measuring the zeta-potential. For ions with higher valency (e.g., Al^3+^ and Ca^2+^ have higher valency than monovalent H^+^, OH^−^, and Na^+^), the zeta-potential decreases in magnitude [47].

### 3.8. In Vitro Sustained Release and Water Dispersibility of BA-Loaded Nanoparticles

Figure 5E shows the in vitro sustained release curves of free BA and BA-loaded zein/G-zein nanoparticles in PBS (pH = 7.4). The slope of the free BA release curve was very large; approximately 45% was released after 1 h, and almost 90% after 4 h, indicating that free BA was quickly released in the simulated environment in vitro. For zein@BA and G-zein@BA, only approximately 30% of BA was released after 4 h, and approximately 90% of it was released after 72 h. Zein@BA and G-zein@BA exhibited obvious sustained-release characteristics. There is a balanced system between BA and hydrophobic proteins, and BA cannot escape without external interference, so the time of drug release is prolonged [48]. However, compared to zein@BA, the sustained release effect of G-zein@BA was slightly improved, which is attributed to the stronger hydrophobic interaction between G-zein and BA.

Free BA was difficult to dissolve in water, with a solubility of only 16 μg/mL (Figure 5F). However, after interacting with protein, the solubility of BA in water increased significantly (*p* < 0.05). The dispersions of zein@BA and G-zein@BA nanoparticles in water were 36.77 and 47.13 μg/mL, respectively, which increased 2.27- and 2.91-fold, respectively. Compared to zein@BA, BA is G-zein@BA more easily dispersed in water. BA crystals were wrapped after the formation of nanoparticles, so the migration of BA molecules was reduced with stronger dispersion in water. In addition, G-zein wrapped BA more closely, so G-zein@BA was more soluble in water [49].

### 3.9. Cytotoxicity Effect of BA-Loaded Nanoparticles in HepG2 Cells

The antitumor activities of free BA, zein@BA, and G-zein@BA on HepG2 cells were assessed using CCK-8 assays. As shown in Figure 6A, the cell survival rate was greater than 90% in the blank nanoparticle group. With increasing concentration and the extension of treatment time, the inhibition of cell proliferation by free BA, zein@BA, and G-zein@BA was enhanced. Compared to that of zein@BA and G-zein@BA, the inhibition of cell proliferation by free BA was stronger after 24 h of treatment. The IC_50_ values of HepG2 cells treated with zein@BA and G-zein@BA for 24 h were 40.98 μg/mL and 35.17 μg/mL, respectively. For treatment with free BA for 24 h, the IC_50_ was 20.85 μg/mL, which was lower than that of zein@BA and G-zein@BA, also indicating that free BA has stronger inhibitory activity on HepG2 cells. The release of BA by BA-loaded nanoparticles is a slow-release process that reduces the accumulated concentration of free BA in cells [50]. These results also showed that only approximately 60% of the BA was released from BA-loaded zein/G-zein nanoparticles after 24 h. After treatment with BA-loaded nanoparticles at concentrations of 160 and 320 μg/mL for 48 h, the cell survival of HepG2 cells was less than 20% (Figure 6B), indicating the antitumor activity of BA-loaded nanoparticles against hepatoma carcinoma cells. Similarly, under hyperthermal conditions, BA-loaded nanoparticles also exhibited enhanced antitumor activity in breast cancer cell lines [51].

## 4. Conclusions

In this study, G-zein was successfully fabricated using a covalent reaction between zein and glucose using the wet heating method. Compared to zein, G-zein has higher solubility, lower hydrophobicity, and a looser protein structure. New glycoprotein production occurred in the α-zein and γ-zein subunits of G-zein. zein@BA and G-zein@BA nanoparticles were prepared using the antisolvent method. When the BA concentration was 1 mg/mL, the encapsulation efficiency of BA-loaded nanoparticles was greater than 80%; the combination of zein and BA was stabilized through static and hydrophobic interactions, and BA-loaded nanoparticles were uniformly spherical. Compared to zein@BA nanoparticles, G-zein@BA had a smaller particle size, higher encapsulation rate, and more stable morphology. The sustained release and solubility of G-zein@BA nanoparticles performed better than those of zein@BA. In addition, BA-loaded nanoparticles markedly inhibited the proliferation of HepG2 tumor cells. This study provides new insights into the structural properties and antitumor activity of BA-loaded G-zein nanoparticles, which form the novel basis for future in vivo application of BA composite nanoparticles for tumor therapy.

## Figures and Tables

**Figure 1 foods-11-02589-f001:**
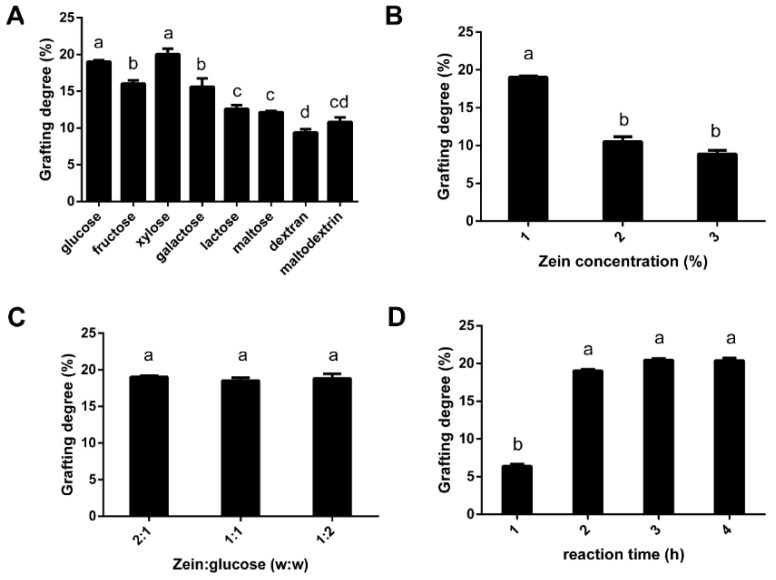
Effects of (**A**) sugar type, (**B**) zein concentration, (**C**) the mass ratio of zein to sugar, and (**D**) reaction time on the grafting degree of zein. Values are presented as the mean ± SD. Different letters represent significant differences (*p* < 0.05) in grafting degree.

**Figure 2 foods-11-02589-f002:**
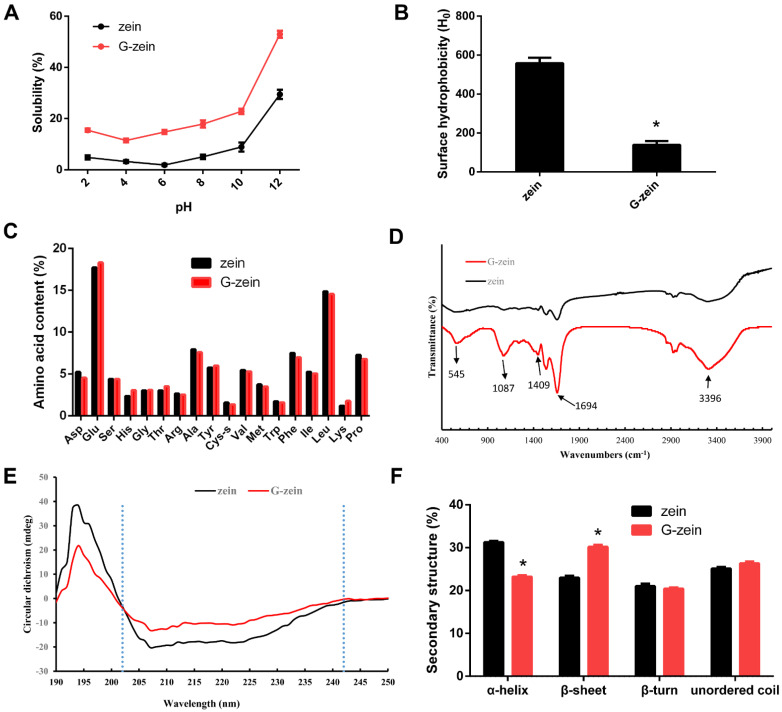
Characterization of zein and G-zein. (**A**) Changes in solubility of zein and G-zein with pH. (**B**) Surface hydrophobicity of zein and G-zein. (**C**) Amino acid composition of zein and G-zein. (**D**) Fourier transform infrared spectroscopy, (**E**) circular dichroism, and (**F**) secondary structure of zein and G-zein. * *p* < 0.05 compared to control.

**Figure 3 foods-11-02589-f003:**
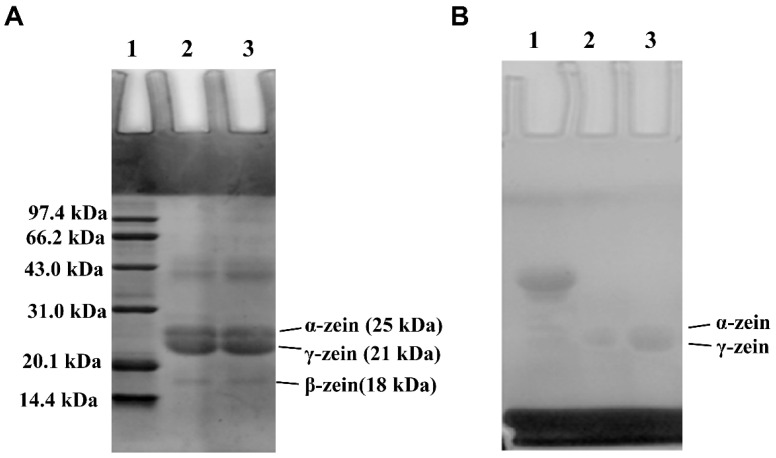
SDS–PAGE of zein and G-zein. (**A**) Coomassie blue staining for protein markers (lane 1), zein (lane 2), and G-zein (lane 3). (**B**) Schiff staining for horseradish peroxidase (lane 1), zein (lane 2), and G-zein (lane 3).

**Figure 4 foods-11-02589-f004:**
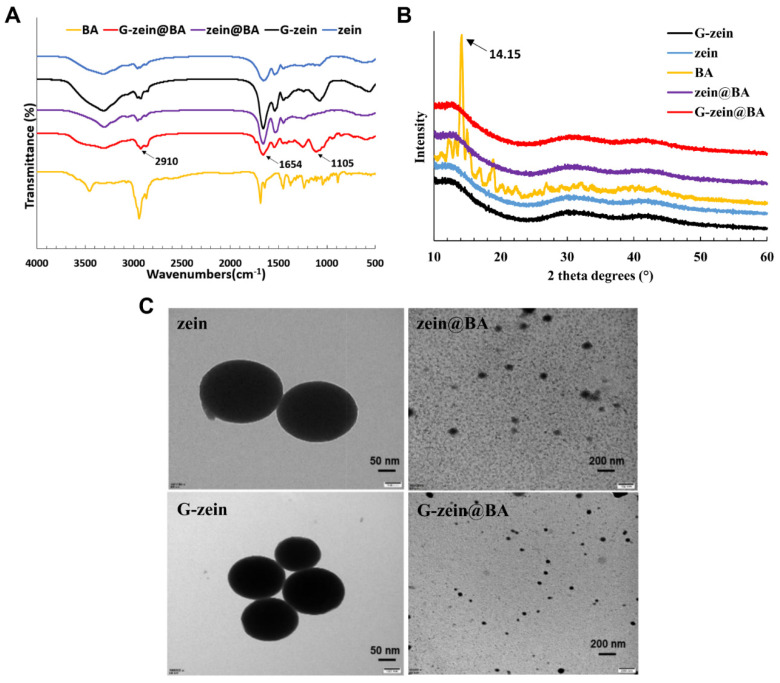
Spectroscopy characteristics of BA-loaded nanoparticles. (**A**) Fourier transform infrared spectroscopy and (**B**) X-ray diffraction patterns of free BA, zein, G-zein, zein@BA, and G-zein@BA. (**C**) Transmission electron microscopy images of zein, G-zein, zein@BA, and G-zein@BA.

**Figure 5 foods-11-02589-f005:**
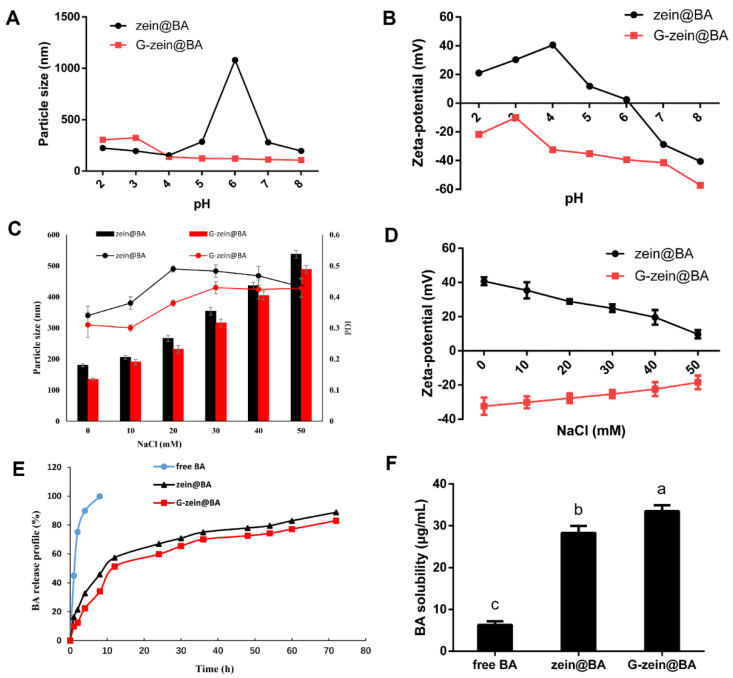
Effects of pH value and ionic strength on BA-loaded nanoparticles. (**A**) Particle size and (**B**) zeta-potential of pH value on zein@BA and G-zein@BA. (**C**) Particle size (column), PDI (line), and (**D**) zeta-potential of NaCl concentration on zein@BA and G-zein@BA. (**E**) In vitro release profiles and (**F**) solubility of free BA, zein@BA, and G-zein@BA. Different letters represent significant differences (*p* < 0.05) in solubility.

**Figure 6 foods-11-02589-f006:**
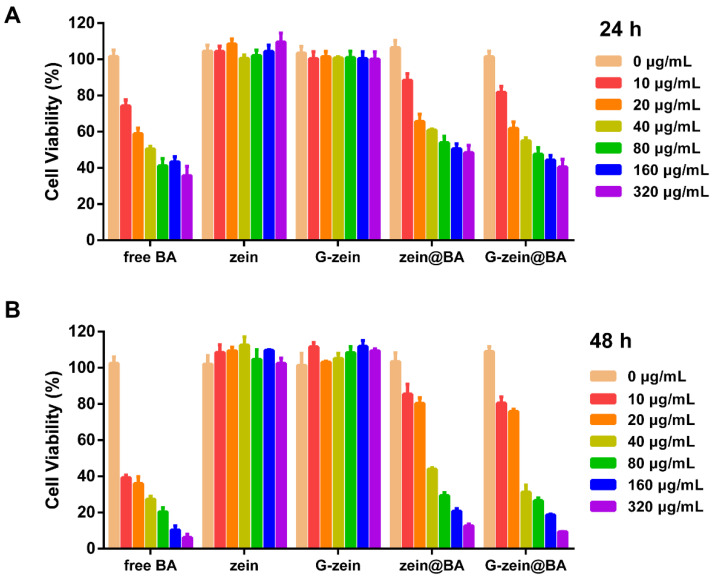
In vitro cytotoxicity of BA-loaded nanoparticles. HepG2 cell viability after free BA, zein, G-zein, zein@BA, or G-zein@BA treatment for (**A**) 24 h and (**B**) 48 h.

**Table 1 foods-11-02589-t001:** The effect of betulinic acid (BA) concentration on zein@BA and G-zein@BA nanoparticle properties.

Sample	BA Concentration (mg/mL)	Particle Size (nm)	Polydispersity Index	Zeta-Potential (mV)	Encapsulation Efficiency (%)
zein	/	130 ± 2.8 ^e^	0.36 ± 0.01 ^e^	30.3 ± 0.78 ^b^	/
G-zein	/	107.9 ± 2.6 ^f^	0.39 ± 0.02 ^de^	−35.3 ± 2.31 ^e^	/
zein@BA	1	168.1 ± 2.2 ^e^	0.48 ± 0.02 ^b^	32.3 ± 0.85 ^b^	83.12 ± 1.18 ^b^
	1.5	274.9 ± 5.6 ^c^	0.45 ± 0.01 ^bc^	32.05 ± 1.63 ^b^	69.61 ± 1.85 ^d^
	2	484.5 ± 5.3 ^a^	0.48 ± 0.02 ^b^	35.1 ± 0.21 ^a^	40.29 ± 0.86 ^f^
G-zein@BA	1	139.3 ± 0.2 ^e^	0.42 ± 0.00 ^cd^	−22.65 ± 1.65 ^c^	89.81 ± 0.79 ^a^
	1.5	223.8 ± 6.0 ^d^	0.54 ± 0.00 ^a^	−22.3 ± 0.85 ^c^	78.24 ± 0.34 ^c^
	2	326.9 ± 1.3 ^b^	0.37 ± 0.00 ^e^	−31.05 ± 0.21 ^d^	49.89 ± 0.35 ^e^

Statistical comparisons were performed using one-way ANOVA with Tukey’s test. Different lowercase letters in each column represent significant differences (*p* < 0.05).

## Data Availability

Data is contained within the article.

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
