# Peer review of "Glycosylated Zein Composite Nanoparticles for Efficient Delivery of Betulinic Acid: Fabrication, Characterization, and In Vitro Release Properties"

_foods, 2022, doi:10.3390/foods11172589_

Round 1
Reviewer 1 Report
Peng et al. have prepared and characterized zein and G-zein nanoparticles for encapsulation of betulinic acid and evaluated their in vitro release and anti-tumor properties. By and large, this paper is well written and the results supported by scientifically sound interpretation. The work described in this study would form the basis for future in vivo application. Therefore, please address the following minor issues:
1. Why there is not a mention of antiproliferative or antitumor activity tested in vitro?
2. Add important quantitative data of the results in the abstract.
3. Line 61, “inhibition on proliferative” should be replaced with “antiproliferative”.
4. All the characteristics described in section 2.3 should be provided with the reference citation for each characteristic for the experimental protocol.
5. Similarly, in sections 2.4 and 2.5, a reference citation should be provided in each of the method description.
6. Figure 4, label the key FTIR peaks and XRD peaks in Fig.4A and 4B.
7. The labels in Figure 4A should enlarged to enhance the clarity and readability.
8. In conclusion, a line or two detailing what would be the future scope of this work will be informative.
Author Response
Response to Reviewer 1 Comments
Dear Reviewer 1:
Thanks for your comments. We have made corrections as suggested. Thank you very much.
Point 1: Peng et al. have prepared and characterized zein and G-zein nanoparticles for encapsulation of betulinic acid and evaluated their in vitro release and anti-tumor properties. By and large, this paper is well written and the results supported by scientifically sound interpretation. The work described in this study would form the basis for future in vivo application. Therefore, please address the following minor issues:
Why there is not a mention of antiproliferative or antitumor activity tested in vitro?
Response 1: Thanks for your comment. In section 3.9. “Cytotoxicity effect of BA-loaded nanoparticles in HepG2 cells”, the antitumor activities of free BA, zein@BA, and G-zein@BA on HepG2 tumor cells were assessed using CCK-8 assays. CCK-8 assay could reflect the effect of BA-loaded nanoparticles on tumor cell proliferation. Please see section 3.9.
Point 2: Add important quantitative data of the results in the abstract.
Response 2: As suggested, we have presented the important quantitative data of the results in the abstract, please see Line 18-19, 22-23.
Point 3: Line 61, “inhibition on proliferative” should be replaced with “antiproliferative”.
Response 3: As suggested, “inhibition on proliferative” has be replaced with “antiproliferative”, please see Line 63-64.
Point 4: All the characteristics described in section 2.3 should be provided with the reference citation for each characteristic for the experimental protocol.
Response 4: As suggested, we have provided with the reference citation for each characteristic for the experimental protocol in section 2.3, please see Line 100 (Reference 14), Line 125 (Reference 15), Line 127 (Reference 16), Line 135 (Reference 17) and Line 140 (Reference 18).
Point 5: Similarly, in sections 2.4 and 2.5, a reference citation should be provided in each of the method description.
Response 5: As suggested, we have provided with the reference citation for each of the method description in section 2.4 and 2.5, please see Line 149 (Reference 19) and Line 159 (Reference 20).
Point 6: Figure 4, label the key FTIR peaks and XRD peaks in Fig.4A and 4B.
Response 6: As suggested, we have label the key FTIR peaks and XRD peaks in Figure 4A and 4B.
Point 7: The labels in Figure 4A should enlarged to enhance the clarity and readability.
Response 7: As suggested, we have enlarged the labels in Figure 4A.
Point 8: In conclusion, a line or two detailing what would be the future scope of this work will be informative.
Response 8: Thanks for your comment. This study provides new insights into the structural properties and antitumor activity of BA composite nanoparticles to aid in the development of zein particles as functional materials to deliver bioactive compounds. The work described in this study would form the basis in vivo application of BA-loaded G-zein nanoparticles for tumor therapy. We have presented the future scope of this work, please see Line 540-543.

Reviewer 2 Report
Dear Editor, Dear Authors,
I had the opportunity to review the paper entitled "Glycosylated zein composite nanoparticles for efficient delivery of betulinic acid: Fabrication, characterization, and in vitro release properties", and I can suggest a minor revision of the paper.
These are my suggestions:
1. Provide the obtained results in the abstract. The current form seems to be like an introduction.
2. Quality of images must be improved.
3. Overall, the organization and interpretation of the result are poor. It should be improved.
4. Novelty of the work should be highlighted in the introduction part as well as the conclusions.
5. All equations need to be numbered.
6. Separate section 2.6. into subsections for more accessible following the mentioned methods
7. Table 1 - indicates the statistical method used for determining statistically significant differences
Author Response
Response to Reviewer 2 Comments
Dear Reviewer 2:
Thanks for your comments. We have made corrections as suggested. Thank you very much.
Point 1: I had the opportunity to review the paper entitled "Glycosylated zein composite nanoparticles for efficient delivery of betulinic acid: Fabrication, characterization, and in vitro release properties", and I can suggest a minor revision of the paper.
Provide the obtained results in the abstract. The current form seems to be like an introduction.
Response 1: Thanks for your comment. As suggested, we have presented the important quantitative data of the results in the abstract, please see Line 18-19, 22-23.
Point 2: Quality of images must be improved.
Response 2: As suggested, we have increased the clarity of all figures.
Point 3: Overall, the organization and interpretation of the result are poor. It should be improved.
Response 3: Thanks for your comment. As suggested, we have revised the organization and interpretation of the result throughout.
Point 4: Novelty of the work should be highlighted in the introduction part as well as the conclusions.
Response 4: Thanks for your comment. Most importantly, compared to zein@BA nanoparticles, G-zein@BA had the advantages of smaller droplets, higher encapsulation efficiency, a more stable morphology, better sustained release and solubility. This study provides new insights into the structural properties and antitumor activity of G-zein@BA, which form the novel basis for future in vivo application for tumor therapy. As suggested, we have highlighted the research novelty in introduction and conclusion, please see Line 64-67, 540-543.
Point 5: All equations need to be numbered.
Response 5: As suggested, we have numbered the equations, please see Line 109, 146, 186 and 237.
Point 6: Separate section 2.6. into subsections for more accessible following the mentioned methods.
Response 6: As suggested, we have separated section 2.6. into subsections (2.6.1-2.6.5).
Point 7: Table 1 - indicates the statistical method used for determining statistically significant differences.
Response 7: Thanks for your comment. As suggested, we have presented the statistical method used for determining statistically significant differences in Table 1, please see Line 372-373.

Reviewer 3 Report
Dear Authors,Re: [Manuscript ID: foods-1861228
Title: "Glycosylated zein composite nanoparticles for efficient delivery of betulinic acid:
Fabrication, characterization, and in vitro release properties"]
In the reported research, you aimed to improve the therapeutic (anticancer) effects
of betulinic acid via encapsulation of the drug using zein nanoparticles. I enjoyed reading
your article and the clear / high quality illustrations.
Please find my comments / suggestions below:
1. In the Abstract you mentioned: "... BA-loaded nanoparticles markedly inhibited ...".
However, you did not specify which formulation (zein or G-zein?);
2. Section 2.1. Materials: majority of reagents / compounds are missing;
3. Section 2.2. Preparation of zein glycosylated products: which "sugar" was used?;
4. Section 2.5. Preparation of BA-loaded zein/G-zein nanoparticles: Temperature of each
steps of the process need to be mentioned. For instance you mentioned: "... and stirred
magnetically at 600 rpm for 2 h." (at what Temperature?);
5. Section 2.6. Characterization of BA-loaded zein/G-zein nanoparticles: can be divided into more sections, e.g. separate sections for Encapsulation Efficiency; Microscopy; X-
Ray; ... etc.;
6. For Particle Sizing what was the Medium Viscosity and Refractive Index parameters?;
7. Particle Sizing: Dilution factor is missing;
8. Encapsulation Efficiency: what was the equation used?;
9. Particle Number evaluation is missing. Consult recently published equation: Sci. Pharm.
2021, 89(2), 15; https://doi.org/10.3390/scipharm89020015
10. Section 2.9. Cytotoxicity of BA-loaded zein/G-zein nanoparticles to HepG2 cells: You
mentioned: "... proliferation was examined using a CCK-8 kit ...". Please provide more
details;
11. Table 1: Data for Control sample (i.e., Empty nanoparticles with no drug) missing.
Please add particle size, ZP, etc. for empty zein nanoparticles;
12. In Table 1, BA concentration (mg/mL). Please double check; was it mg or microgram?;
13. In Lines 359-360 you mentioned: "... With increasing BA concentration, the
encapsulation efficiency of BA in zein@BA and 359 G-zein@BA decreased ...". Please
explain the reason behind this observation;
14. In Figure 5.A. particle size of zein@BA sample suddenly increases. I was not able to
find an explanation for this finding;
15. Adequate Control sample data are missing in Figure 5 (e.g. empty nanoparticles). This
is while Figure 6 data sets are complete and meaningful;
16. For Discussion of particle size, PDI and ZP variations observed in this research, consult
the following 2 comprehensive manuscripts:
DOI: 10.3390/pharmaceutics10020057
https://www.mdpi.com/1999-4923/10/2/57
and:
https://doi.org/10.1016/j.heliyon.2018.e01088 and https://doi.org/10.1016/j.jconrel.2016.06.017
17. Please make the Aim of the research more clear in the Conclusion. Also, distinguish
between your work with previously published papers. Check the following similar publications: https://doi.org/10.1016/j.cis.2017.06.012 and https://doi.org/10.1039/C2CS15362A and doi: 10.1097/CJI.0b013e318234ecf5
Author Response
Response to Reviewer 3 Comments
Dear Reviewer 3:
Thanks for your comments. We have made corrections as suggested. Thank you very much.
Point 1: In the reported research, you aimed to improve the therapeutic (anticancer) effects of betulinic acid via encapsulation of the drug using zein nanoparticles. I enjoyed reading your article and the clear / high quality illustrations.
Please find my comments / suggestions below:
In the Abstract you mentioned: "... BA-loaded nanoparticles markedly inhibited ...". However, you did not specify which formulation (zein or G-zein?);
Response 1: Thanks for your comment. Both zein@BA and G-zein@BA nanoparticles could markedly inhibit the proliferation of HepG2 cells. We have revised the sentence, please see Line 23-24.
Point 2: Section 2.1. Materials: majority of reagents / compounds are missing;
Response 2: As suggested, we have added the reagents and compounds in section 2.1, please see Line 74-81.
Point 3: Section 2.2. Preparation of zein glycosylated products: which "sugar" was used?;
Response 3: In section 2.2, sugar type used in this study contains monosaccharides, disaccharides and polysaccharides (glucose, fructose, xylose, galactose, lactose, maltose, dextran and maltodextrin), please see Line 94-95.
Point 4: Section 2.5. Preparation of BA-loaded zein/G-zein nanoparticles: Temperature of each steps of the process need to be mentioned. For instance you mentioned: "... and stirred magnetically at 600 rpm for 2 h." (at what Temperature?);
Response 4: Zein and G-zein (2 g) were dissolved in 70% ethanol (v/v, 100 mL) and stirred magneti-cally at 600 rpm for 2 h at 25 °C. As suggested, we have presented with the temperature of each steps of the process in section 2.5, please see Line 159-163.
Point 5: Section 2.6. Characterization of BA-loaded zein/G-zein nanoparticles: can be divided into more sections, e.g. separate sections for Encapsulation Efficiency; Microscopy; X-Ray; ... etc.;
Response 5: As suggested, we have separated section 2.6. into subsections (2.6.1. Particle size, polydispersity index (PDI), and zeta potential; 2.6.2. Encapsulation efficiency; 2.6.3. FTIR; 2.6.4. transmission electron microscopy (TEM); 2.6.5. X-ray diffractometer).
Point 6: For Particle Sizing what was the Medium Viscosity and Refractive Index parameters?;
Response 6: For particle sizing, the Medium Viscosity is 0.8872, and the Refractive Index is 1.33, please see Line 171.
Point 7: Particle Sizing: Dilution factor is missing.
Response 7: For particle sizing, the dilution factor is 100-fold. As suggested, we have presented the dilution factor, please see Line 172.
Point 8: Encapsulation Efficiency: what was the equation used?
Response 8: Thanks for your comment. The encapsulation efficiency was the mass ratio of BA-loaded nanoparticles to total BA. As suggested, we have presented the equation used for encapsulation efficiency, please see Line 186.
Point 9: Particle Number evaluation is missing. Consult recently published equation: Sci. Pharm. 2021, 89(2), 15; https://doi.org/10.3390/scipharm89020015
Response 9: Thanks for your suggestion. We have referred to the paper you provided. The cumulative mean diameter was measured by Stokes-einstein equation. The zeta potential was measured in Smoluchowski mode at 25 °C. Please see Line 173-174 (reference 21).
Point 10: Section 2.9. Cytotoxicity of BA-loaded zein/G-zein nanoparticles to HepG2 cells: You mentioned: "... proliferation was examined using a CCK-8 kit ...". Please provide more details;
Response 10: Thanks for your comment. As suggested, we have presented the formula about CCK-8 assay, please see Line 231-237.
Point 11: Table 1: Data for Control sample (i.e., Empty nanoparticles with no drug) missing. Please add particle size, ZP, etc. for empty zein nanoparticles;
Response 11: As suggested, we have added the data for control sample (empty nanoparticles with no drug), please see Table 1.
Point 12: In Table 1, BA concentration (mg/mL). Please double check; was it mg or microgram?;
Response 12: Thanks for your kind reminder. We have confirmed that the concentration unit of BA is mg/mL.
Point 13: In Lines 359-360 you mentioned: "... With increasing BA concentration, the
encapsulation efficiency of BA in zein@BA and G-zein@BA decreased ...". Please explain the reason behind this observation;
Response 13: Thanks for your comment. BA bound to the hydrophobic microregion of protein through electrostatic and hydrophobic interaction, so that it could be effectively encapsulated in nanoparticles. With the increase of BA concentration, the hydrophobic binding site of zein/G-zein could not meet more BA binding, so the encapsulation efficiency decreased. The interaction force between BA and protein was limited, and as the concentration of BA increases, the protein cannot produce enough electrostatic or hydrophobic interactions with BA, resulting in particle aggregation and reduced encapsulation efficiency (Dai, Li, Wei, Sun, Mao & Gao, 2018). We have explained the reason behind this observation, please see Line 395-401.
Point 14: In Figure 5.A. particle size of zein@BA sample suddenly increases. I was not able to find an explanation for this finding;
Response 14: Thanks for your comment. In Figure 5A, particle size of zein@BA sample suddenly increases at pH=6. Because the isoelectric point of zein was at approximately pH=6, protein aggregation and precipitation occurred under this pH value condition, and the particle size of zein@BA nanoparticles increased suddenly (Shah, Xu & Mraz, 2021). As suggested, we have presented the explanation for this finding, please see Line 447-449.
Point 15: Adequate Control sample data are missing in Figure 5 (e.g. empty nanoparticles). This is while Figure 6 data sets are complete and meaningful;
Response 15: Thanks for your suggestion. In Figure 5, we focus on comparing the differences between zein@BA and G-zein@BA nanoparticles, ignoring control samples (empty nanoparticles). The reviewer gave us the exciting suggestion for the future studies, which need systematic and complete experimental design and research.
Point 16: For Discussion of particle size, PDI and ZP variations observed in this research, consult the following 2 comprehensive manuscripts:
DOI: 10.3390/pharmaceutics10020057 https://www.mdpi.com/1999-4923/10/2/57
and: https://doi.org/10.1016/j.heliyon.2018.e01088
and https://doi.org/10.1016/j.jconrel.2016.06.017
Response 16: Thanks for your comment. As suggested, we have revised the discussion of particle size, PDI and ZP variations by referring to the papers you provided, please see Section 3.5. Line 360-362 (reference 33), 364-367 (reference 34), 384-388 (reference 35) and Section 3.7. Line 456-459 (reference 43), 476-478 (reference 45).
Point 17: Please make the Aim of the research more clear in the Conclusion. Also, distinguish between your work with previously published papers. Check the following similar publications: https://doi.org/10.1016/j.cis.2017.06.012
and https://doi.org/10.1039/C2CS15362A
and doi: 10.1097/CJI.0b013e318234ecf5
Response 17: Thanks for your comment. This study provides new insights into the structural properties and antitumor activity of BA-loaded G-zein nanoparticles, which form the novel basis for future in vivo application of BA composite nanoparticles for tumor therapy. As suggested, we have revised the conclusion by referring to the papers you provided, please see Line 540-543.
References:
Dai, L., Li, R., Wei, Y., Sun, C., Mao, L., & Gao, Y. (2018). Fabrication of zein and rhamnolipid complex nanoparticles to enhance the stability and in vitro release of curcumin. Food Hydrocolloids, 77, 617-628.
Shah, B. R., Xu, W., & Mraz, J. (2021). Formulation and characterization of zein/chitosan complex particles stabilized Pickering emulsion with the encapsulation and delivery of vitamin D-3. Journal of the Science of Food and Agriculture, 101(13), 5419-5428.
